# Sphingosine 1-Phosphate Signaling at the Skin Barrier Interface

**DOI:** 10.3390/biology11060809

**Published:** 2022-05-25

**Authors:** Kana Masuda-Kuroki, Anna Di Nardo

**Affiliations:** Department of Dermatology, School of Medicine, University of California, San Diego, CA 92093, USA; kkuroki@health.ucsd.edu

**Keywords:** sphingosine 1-phosphate, sphingosine 1-phosphate receptors, keratinocytes, mast cells

## Abstract

**Simple Summary:**

Sphingosine 1-phosphate is a bioactive phospholipid derived from cell membranes. It is biosynthesized and metabolized within the cell and released extracellularly to trigger various cellular responses via sphingosine 1-phosphate receptors. Keratinocyte and mast cell functions are responsible for skin barrier acquired and innate immunity, and express sphingosine 1-phosphate receptors. These receptors regulate not only cell differentiation and proliferation, but also immune responses such as cell migration and cytokine secretion, through G-protein-mediated signaling pathways. Sphingosine 1-phosphate and its receptor signaling pathways are also involved in several skin diseases along with various receptor agonists and are being investigated as potential therapeutic agents. Here, we focus on the association of sphingosine 1-phosphate and its receptors within the skin barrier interface.

**Abstract:**

Sphingosine 1-phosphate (S1P) is a product of membrane sphingolipid metabolism. S1P is secreted and acts via G-protein-coupled receptors, S1PR1-5, and is involved in diverse cellular functions, including cell proliferation, immune suppression, and cardiovascular functions. Recent studies have shown that the effects of S1P signaling are extended further by coupling the different S1P receptors and their respective downstream signaling pathways. Our group has recently reported that S1P inhibits cell proliferation and induces differentiation in human keratinocytes. There is a growing understanding of the connection between S1P signaling, skin barrier function, and skin diseases. For example, the activation of S1PR1 and S1PR2 during bacterial invasion regulates the synthesis of inflammatory cytokines in human keratinocytes. Moreover, S1P-S1PR2 signaling is involved in the production of inflammatory cytokines and can be triggered by epidermal mechanical stress and bacterial invasion. This review highlights how S1P affects human keratinocyte proliferation, differentiation, immunoreaction, and mast cell immune response, in addition to its effects on the skin barrier interface. Finally, studies targeting S1P-S1PR signaling involved in inflammatory skin diseases are also presented.

## 1. Introduction

Sphingosine 1-phosphate (S1P) is a bioactive sphingolipid which modulates cell proliferation and differentiation, including angiogenesis and apoptosis [1,2,3].

S1P generation and secretion play a vital role in many intracellular signaling cascades and pathogeneses. In particular, angiogenesis, cell proliferation and differentiation, as well as immune and inflammatory responses, have been widely investigated. It also plays an essential role in signaling molecules, immune cell trafficking, and in various diseases in multiple organs by binding to one of five G-protein-coupled receptors, S1PR1-5 [4].

The functions of S1P and the S1P-S1PR signaling pathway are equally important in the skin organ. They are involved in keratinocyte differentiation and proliferation [5,6], mast cell degranulation and migration [7,8], and have been shown to contribute to the pathogenesis of skin sclerosis, psoriasis, and atopic dermatitis, as well as in the defense of bacterial infections [9,10,11,12,13,14].

We begin with the recent accumulated research on the effects of S1P and S1P-S1PR signaling on the skin, as it is important to consolidate common findings, in order to advance in this growing field.

## 2. S1P Formulation and Secretion

Until the early 1990s, S1P was thought to be an intermediate metabolite produced during the degradation of sphingomyelin and sphingolipids, which are abundant in the cell membrane. Subsequently, S1P was found to be an up-regulator of fibroblasts [3]; and also involved in cell proliferation, cell motility, morphology modulation, and cell differentiation, including tumor cells, neurons, vascular smooth muscle, and vascular endothelial cells [15].

S1P is derived from the cell membrane sphingolipids. Sphingomyelin in the cell membrane is cleaved by sphingomyelinase to yield ceramide and is then cleaved by ceramidase to form sphingosine. Sphingosine can be phosphorylated by sphingosine kinases (SphK1 and SphK2), leading to the formation of S1P [2,3,15]. S1P is formed in the cells but is degraded by intracellular S1P lyase or dephosphorylated by S1P phosphatases (SPP) [2,15,16,17] (Figure 1).

Plasma S1P concentrations are high in the μM order [18], but most of it is bound to albumin and high-density lipoprotein (HDL), resulting in low levels of free S1P (less than 2%) [19]. In contrast, the concentration of S1P in the stromal fluid is considered to be in the nM order [20]. Erythrocytes and endothelial cells produce S1P constantly [21], while other cells, such as mast cells and keratinocytes, produce and release S1P in response to activation [22,23]. For example, Olivera et al. [22] reported that antigen administered to mouse models of anaphylaxis resulted in increased S1P production from mast cells along with increased histamine levels. Park et al. [23] reported that the exposure of keratinocytes to low toxic doses of Tg, a specific pharmacological ER stressor, increases intracellular ceramide levels and its distal metabolites, sphingosine and S1P. Moreover, our group previously reported that stimulating keratinocytes with Staphylococcus aureus culture supernatant resulted in a threefold increase in S1P production [13].

## 3. S1P Receptors

In the late 1990s, a family of five G-protein-coupled receptors (GPCR) for S1P, S1PR1-5, were identified [24,25,26,27,28]. It was first shown that S1P induces the activation of extracellular-signal-regulated kinase (ERK) 1 and 2 via activation of Gi proteins [29]. Subsequently, the GPCR, initially called endothelial differentiation gene 1 (EDG1), was identified as the receptor for S1P [30]. The binding of the extracellular S1P to S1PRs on the cell membrane results in signaling and cellular responses, such as cell proliferation, differentiation, apoptosis, and immunoreaction. S1PR1-3 is found in high density in the cardiovascular and immune cells and is widely distributed in most other cell types. On the other hand, S1PR4 and S1PR5 can also have a limited distribution, such as in lymphatic and nervous cells [31,32].

Several groups have studied the signaling mechanisms of S1P receptors in detail. S1PR1 is mainly bound to trimeric G protein Gi/o and activates ERK via activation of low-molecular-weight G protein Ras and activates phosphoinositide 3-kinase-mediated activation of Akt and low-molecular-weight G protein Rac, phospholipase C (PLC), and inhibition of cyclic AMP production [28,33]. S1PR2 mainly activates the low-molecular-weight G protein Rho via G12/13, and Rho activates Rho kinase (ROCK/ROK). S1PR2 also inhibits cell migration by suppressing Rac downstream of Rho without ROCK [34]. Besides G12/13, S1PR2 also binds to Gi/o and Gq [35]. S1PR3 primarily activates PLC by conjugating Gq, causing Ca2+ mobilization, and activating C kinase (PKC) [36]. S1PR3 binds to Gi/o and G12/13 and activates the downstream Rho pathways [37]. S1PR4 binds to Gi/o and G12/13 and activates ERK, mitogen-activated protein kinase (MAPK), and the PLC downstream pathways [38]. S1PR5 binds to Gi/o and G12/13 and promotes the downstream signaling pathways [35] (Figure 2).

## 4. S1P and S1P-S1PR Pathways on Human Keratinocytes

Since the first discovery of S1P, the effects of S1P on keratinocytes have also been documented. Although S1P has a growth-promoting effect in many cells, keratinocyte S1P exposure causes cell growth arrest. Vogler et al. [39] reported that both intracellular actions of S1P itself and S1PRs cause cell growth arrest. Kim et al. [40] reported that S1P plays an essential role in the negative regulation of keratinocyte proliferation by inhibiting the Akt/PKB pathway.

Besides inhibiting proliferation, there is also a substantial increase in intracellular calcium content due to S1P [41]. Since this is the most critical signal of the differentiation process in keratinocytes, they are transformed into corneocytes [11].

S1P can also protect keratinocytes from their apoptosis. Uchida et al. [42] reported that the conversion of sphingosine to S1P protects keratinocytes from UVB-induced, ceramide-mediated apoptosis in keratinocytes. More et al. [43] reported that S1P attenuates hydrogen-peroxide-induced apoptosis of keratinocytes by promoting phosphorylation of the Akt pathway.

The immunomodulating effects of S1P are well known as central in regulating the circulation of lymphocytes, plasma, and tissue-to-tissue T lymphocytes [18,20,21], but S1P also exhibits immunomodulating effects in the skin, particularly in keratinocytes. Park et al. [44] reported that increased S1P production induced by ER stress activates NFκB- and C/EBPα-dependent pathways and promotes keratinocyte CAMP production. It has also been observed that S1P binds to heat shock proteins after ER stress [45]. In addition, activation of SPHK1 promotes cathelicidin production by keratinocytes, while SPHK2 has the opposite role, or is not associated at all [46,47]. Furthermore, S1P stimulates the production of inflammatory cytokines, such as TNF-α, IL-8, and IL-36gamma from human keratinocytes [13,48].

S1P is also known to promote skin wound healing by increasing keratinocyte migration [49]. For example, Shin et al. [50] reported that a ginsenoside species, Rb1, which regulates sphingolipid metabolism, promotes keratinocyte migration using the S1P-S1P receptor–ERK1/2-NF-kB-MMP2/9 pathway.

Epidermal keratinocytes express all five types of S1PR 1-5 [39], most abundantly S1PR5, followed by S1PR1 and S1PR2 [13]. S1P potently increases Ca (2+), which plays an essential role in keratinocyte differentiation, and S1PR1 mediates chemotaxis of keratinocytes with no effect on increasing Ca (2+), though S1PR3 increases S1P-induced Ca (2+) [41]. S1P inhibits keratinocyte proliferation; as also mentioned above, S1PR2 was found to be dominantly involved in the S1P-induced dephosphorylation of Akt and keratinocyte growth arrest [51]. Schmitz et al. [52] reported that S1PR3 in human keratinocytes mediates eNOS activation and NO- formation in response to S1P. They also reported that the S1PR3 deficiency inhibits S1P to protect human keratinocytes from apoptosis [52].

S1P receptors have been found to be associated not only with keratinocyte proliferation, differentiation, and apoptosis control but also with immunoreaction and inflammatory mechanisms. It has been shown that S1P-induced increases in TNFα, IL-8, and IL-36γ are mediated by S1PR1 and 2, and that the above cytokine increases from Staphylococcus-aureus-stimulated keratinocytes are mediated by S1PR1 [13]. Furthermore, S1PR2 expression is upregulated in Staphylococcus-aureus-stimulated keratinocytes, and S1PR2 is involved in the formation of skin barrier structures, such as filaggrin, claudin, occludin, and corneodesmosin [13,14]. Moreover, S1PR3 has recently been reported to contribute to the upregulation of IL-17A and IL-22 secretion via the AKT/mTOR pathway [53] (Figure 3).

## 5. S1P and S1P-S1PR Pathways on Human Mast Cells

As mentioned above, S1P is involved not only in cell proliferation, migration, and differentiation, but also in immune and inflammatory responses. Therefore, considering the relationship between mast cells and S1P, a significant player in inflammatory and allergic immunoreactions, is key for understanding skin inflammation and has been investigated by many groups.

Mast cells release S1P in response to IgE or antigen stimulation [7] because cross-linking of FcɛRI increases SphK activity and the production of S1P [54]. Fyn kinase, a 59 kDa member of the Src family of tyrosine kinases [55], is required for this reaction [56]. It has also been reported that multidrug-resistant protein 1 (MRP1) contributes to releasing S1P from the intracellular to the extracellular space of mast cells [57]. SPHK1 and 2 are activated by antigen stimulation; while SPHK1 is a critical factor in regulating the human mast cell response, SPHK2 is essential in mouse mast cell response [58]. Mast cells release S1P, but on the other hand, S1P stimulates mast cell chemotaxis, degranulation, and cytokine secretion. Jolly et al. [8] reported that even a very low concentration of S1P (1 nM) enhances mast cell chemotaxis, and 100 nM S1P induces cytokine secretion, such as TNFα, IL-13, IL-6, and MCP-1, from mast cells by using bone-marrow-derived mast cells and RBL-2H3 cells. Oskeritzian et al. [59] also reported that S1P stimulates mast cell cytokine and chemokine secretion, such as TNFα, IL-6, and CCL2/MCP-1, by using LAD2 mast cells and cord-blood-derived human mast cells. Interestingly, S1P not only affects the inflammatory response of mast cells but also regulates their differentiation and maturation. For example, Price et al. [60] reported that S1P induces the maturation of cord-blood-derived mast cells to express chymase. Olivera et al. [61] reported that chronic exposure to S1P on mast cells enhances the expression of genes associated with calcium response and degranulation.

Mast cells express S1PR1, S1PR2, and S1PR4, but not S1PR3 and S1PR5 [8,54,62,63].

S1P released from mast cells enhances mast cell migration and the secretion of various allergenic and inflammatory mediators in an autocrine manner by binding to S1PR1 and S1PR2 [54,64,65]. For example, S1PR1 regulates mast cell migration toward low concentrations of antigen, while the migration of mast cells is inhibited through the S1P–S1PR2 axis [54,64]. However, once the mast cell reaches the target site, degranulation is induced by the S1P-S1PR2 pathway [65]. S1PR4 is a negative regulator of IgE-induced degranulation via IL-33, although it has little effect on mast cell migration [63]. S1PR4 is also involved in mast cell degranulation and cytokine production [63]. As proof of this, Jeon et al. [66] reported that the administration of CYM50358, a selective S1PR4 antagonist, suppressed degranulation of RBL-2H3 mast cells and reduced IL-4 production and serum IgE levels among ovalbumin-induced allergic model mice (Figure 4).

## 6. S1P-S1PR Signaling and Skin Diseases

Although many effects of S1P signaling in the human organism have been discovered, the full extent of its functions has not been elucidated yet. It is now becoming clear that dysfunction or imbalance of the S1P axis is a contributing factor in inflammatory skin diseases such as atopic dermatitis and psoriasis [67].

For atopic dermatitis, in a mouse model, Japtok et al. [68] applied topical S1P and observed reduced antigen uptake capacity by epidermal dendritic cells. They described the reason for this; the Akt signaling pathway is inhibited in dendritic cells and in keratinocytes. Park et al. [69] reported that, in a mouse model of atopic dermatitis, the administration of JTE-013, a selective antagonist of S1PR2, followed by a topical application of 2,4-dinitrochlorobenzene, decreased lymph node size and levels of inflammatory cytokines such as IL-4, IL-13, IL-17, and IFN-γ in the ear and lymph nodes, and levels of chemokines CCL17 and CCL22. They suggested that this is because the expression of CCL17 and CCL22 induced by IL-4 is significantly blunted in bone-marrow-derived dendritic cells (BMDCs) from S1pr2-gene-deficient mice and that JTE-013 suppresses this induction in BMDCs from wild-type mice. They also observed that the inflammation of atopic dermatitis is significantly improved in S1pr2-gene-deficient mice.

In the imiquimod model, Schaper et al. [9] reported that blocking S1P significantly reduced swelling, inflammatory cell infiltration, and edema in the ear skin for psoriasis-like skin lesions. Syed et al. [70] reported that bone-marrow-specific S1PR1 deficiency promotes early inflammation in imiquimod-induced psoriasiform dermatitis mouse models because the S1PR1 deletion alters IL-1β and VEGF generation and expression of their receptors, affecting neoangiogenesis and neolymphangiogenesis. Jena et al. [71] reported that the Western diet increases the incidence of dermatitis in mice and S1PR2 expression and genes encoding SPHKs, S1P phosphatase, binding proteins, and transporters are elevated in skin lesions. S1PR3 is also related to the mechanism of psoriasis. He et al. [53] reported S1PR3 axis contributes to the hyperproliferation of keratinocytes and skin inflammation in psoriasis via the AKT/mTOR pathway. For S1PR4, Schuster et al. [72] reported that imiquimod-induced psoriatic skin lesions created in S1pr4-deficient mice reduce the severity of psoriasiform dermatitis, decrease CCL2, IL-6, and CXCL1, and reduce infiltrating monocytes and granulocytes. This conclusion is drawn because S1PR4 signaling is associated with TLR signaling in macrophages to produce CCL2 via the NF-κB pathway.

Furthermore, S1P-S1PR signaling also has a role in the retention of resident memory T cells (T_RM_), which is essential for the pathogenesis of chronic inflammatory and autoimmune diseases as well as for protection against pathogens. Skon et al. [73] reported that accumulation of activated T_RM_ cells in inflamed skin is dependent on the downregulation of S1PR1. Evrard et al. [74] reported that downregulation of S1PR5 resulted in the retention of T_RM_ cell precursors in the skin and provided skin with immune protection.

## 7. S1P and S1PR Modulators for Skin Diseases

Beginning with the development of Fingolimod (FTY720) for the treatment of multiple sclerosis [75], various S1P and S1P receptor modulators have been developed. Given that S1P and S1P-S1PR signaling have been implicated in the pathogenesis of skin diseases, a variety of modulators have been investigated in an effort to develop therapeutic agents.

FTY720 is a functional antagonist of S1PR1 and has been documented to be involved in S1PR2, S1PR3, S1PR4, and S1PR5 [76,77].

Administration of FTY720 into mouse skin, after repeated hapten administration, inhibits eosinophil outflow from the bone marrow and reduces the number of eosinophils in the blood and skin [78]. Thus, FTY720, in combination with betamethasone ointment applied to an NC/Nga mouse model of spontaneous development of atopic dermatitis-like skin inflammation, inhibits epidermal hyperplasia and mast cell accumulation [79]. The systemic application of FTY720 also protected NC/Nga mice from skin inflammation [80]. Additionally, using histamine- and IgE-induced dermatitis mouse models, topical application of FTY720, as a preventive measure or after the establishment of inflammation has been shown to reduce SPHK1 activity and cytokine production, such as TNFα, IL-6, and MCP-1, and reduces ear thickness [81]. Furthermore, FTY720 inhibited the migration of dendritic cells from the skin to the lymph nodes [82]. In addition to these reports, it has also been shown that FTY720-containing gel decreases the Eczema Area and Severity Index (EASI) score of AD-like lesions in SKH-1 hairless mice [83]. FTY720 is also effective in psoriasis. In the IMQ-induced psoriasiform dermatitis mouse model, FTY720 inhibited the migration of IL-17A-producing γδT cells from lymph nodes to the skin [84,85]. Ramírez et al. [86] reported that FTY720 treatment in IMQ-induced psoriasiform dermatitis mouse models showed an accumulation of Vγ4 + Vδ4+ cells in the responding LNs and inhibited their increase both in the blood and in the inflamed skin. Moreover, FTY720 inhibits hypertrophic scar fibroblasts progression and the G0/G1 cycle and promotes apoptosis through S1PR5 [87].

The selective S1P1 modulator Syl930 reduces sodium lauryl sulfate (SLS)-induced psoriasis-like skin lesions in mouse models [88]. In addition, Syl930 improved skin lesions in a propranolol-induced psoriasis-like skin lesion guinea pig model [88]. In a phase II study of psoriasis, the oral administration of the S1PR1 agonist ponesimod reduced the psoriasis area and severity index (PASI) score by 75% in 77% of patients [89]. Unfortunately, systemic side effects such as lymphopenia and transient bradycardia were observed. Therefore, the development of a topical soft-drug S1PR1 agonist modified from ponesimod has been attempted [90,91]. IMMH002, a recently developed oral S1PR1 modulator, reduces irritation in the SDS-induced psoriasis-like skin lesion mouse model, T lymphocyte infiltration of the imiquimod-induced psoriasiform dermatitis mouse model, and skin damage in the propranolol-induced psoriasis-like skin lesion guinea pig model [92]. Another S1PR1 modulator, cenerimod, reduces skin and lung fibrosis in a bleomycin-induced SSc mouse model [12].

There have been many trials with S1PR1 modulators in the field of skin diseases, but there have also been studies with other modulators related to the S1P axis. For example, Shin et al. [93] reported that topical application of HWG-35D, an SPHK2 inhibitor, improves skin lesions, serum IL-17A levels, and mRNA levels of IL-17A, K6, and K16 genes in imiquimod-induced psoriasiform dermatitis mouse model skin.

## 8. Conclusions

Since the discovery of S1P and its receptors, its association with many vascular, immune, and inflammatory diseases has now become clear. Although further studies are still needed before S1P-S1PR modulators can be applied clinically to skin diseases, it is clear that the S1P-S1PR axis is a key factor within the immunomodulatory mechanism and barrier functions of the skin. With future research, the development of therapeutic reagents which can be applied to skin diseases is expected in the near future.

## Figures and Tables

**Figure 1 biology-11-00809-f001:**
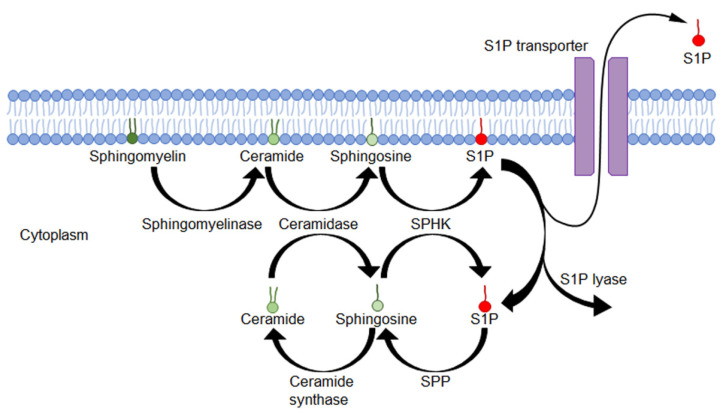
Biosynthesis and metabolic pathways of sphingolipids. Sphingomyelin in the lipid bilayer is converted to ceramide by sphingomyelinase. Ceramide is transformed into sphingosine by ceramidase. Some sphingosines are phosphorylated by sphingosine kinase (SPHK) and converted to the bioactive lipid S1P. S1P is either released in extracellular space via S1P transporters, degraded by S1P lyase, or transformed by sphingosine phosphatase (SPP) to sphingosine.

**Figure 2 biology-11-00809-f002:**
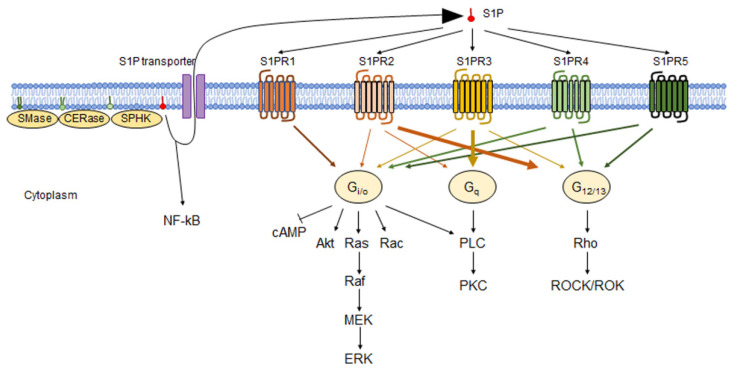
S1P-S1PR pathways. Extracellular S1P binds to S1P receptors. Each receptor binds to a different G protein and triggers downstream signaling pathways. S1PR1 mainly binds to Gi/o, and S1PR2 binds G12/13 and Gi/o and Gq. S1PR3 binds to mostly Gq but also to Gi/o and G12/13. S1PR4 and S1PR5 bind to Gi/o and G12/13. Gi/o activates Akt, Ras-Raf-MEK-ERK, Rac, and PLC/PKC pathways but suppresses cAMP. Gq activates the PLC/PKC pathway. G12/13 activates Rho and the downstream ROCK/ROK pathway.

**Figure 3 biology-11-00809-f003:**
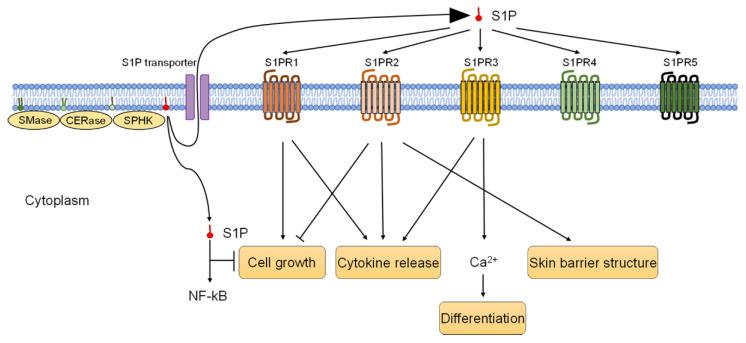
S1P and S1P-S1PR pathways of keratinocytes. Keratinocytes express all five S1P receptors, S1PR1–S1PR5. While S1PR1 activates cell growth, S1PR2 inhibits keratinocyte cell growth and mediates skin barrier structure. S1PR1, S1PR2, and S1PR3 trigger cytokine release. S1PR3 promotes keratinocyte differentiation.

**Figure 4 biology-11-00809-f004:**
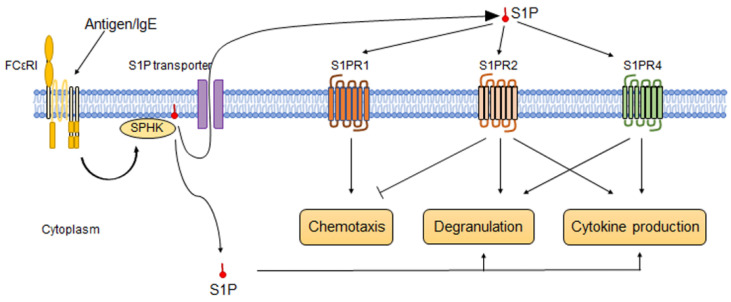
S1P and S1P-S1PR pathways of mast cells. Recognition of antigens via IgE-FcεRI, the plasma membrane of mast cells, activates SPHK, thereby promoting the generation of S1P. The generated S1P stimulates degranulation and cytokine production intracellularly. On the other hand, S1P released into the extracellular space also acts in an autocrine manner via S1P receptors expressed on mast cells. Mast cells express three S1P receptors, S1PR1, S1PR2, and S1PR4, but not S1PR3 and S1PR5. S1PR1 activates mast cell chemotaxis; meanwhile, S1PR2 suppresses it. S1PR2 and S1PR3 promote mast cell degranulation and cytokine production.

## Data Availability

Not applicable.

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
