# Peer review of "Sphingosine 1-Phosphate Signaling at the Skin Barrier Interface"

_biology, 2022, doi:10.3390/biology11060809_

Round 1
Reviewer 1 Report
This is an interesting review on S1P and its receptors, in the skin. I have a few comments:
- The authors could shorten the intro, line 41->49 could go, and at the end of the intro there could be a sum up of the upcoming topics that will follow "first we'll expose...."
- The authors could consider having one section per cell type instead of two (f.i. one for keratinocytes, one for mast cells) and have a schematic for each one summarizing the content of the section. It would increase the readability and the interest of the paper.
- When mentioning psoriasis mouse models, please use psoriasiform, or psoriasis-like instead of psoriasis.
- FTY720 is widely use as an egress blockers for T cells, from the lymph nodes, I think this should be stated a bit more than only for ref 82. Also, there is (at least one) paper showing the role of S1PR5 in the retention of skin tissue resident memory T cells, it would be good to add it, and have a little paragraph on TRM, I suspect your readers would expect that to be mentioned. doi 10.1084, 2021 in JEM.
Author Response
Reviewer: 1
Comments and suggestions for Authors
This is an interesting review on S1P and its receptors, in the skin. I have a few comments:
The authors could shorten the intro, line 41->49 could go, and at the end of the intro there could be a sum up of the upcoming topics that will follow "first we'll expose...."
Response:
Thank you for your observations. According to your suggestion, we have shortened the intro, lines 41 – 49.
The authors could consider having one section per cell type instead of two (f.i. one for keratinocytes, one for mast cells) and have a schematic for each one summarizing the content of the section. It would increase the readability and interest of the paper.
Response:
Thank you for your suggestion. We have combined the sections per cell type and added a diagram for keratinocytes, as shown in Figure 3. In addition, the chart regarding mast cells has been moved to Figure 4.
When mentioning psoriasis mouse models, please use psoriasiform, or psoriasis-like instead of psoriasis.
Response:
Thank you for pointing it out. We corrected the terms for the psoriasis models. We used psoriasiform for the imiquimod-induced psoriasis models and psoriasis-like for other chemicals-induced models.
FTY720 is widely used as an egress blockers for T cells from the lymph nodes; I think this should be stated a bit more than only for ref 82. Also, there is (at least one) paper showing the role of S1PR5 in the retention of skin tissue resident memory T cells, it would be good to add it, and have a little paragraph on TRM, I suspect your readers would expect that to be mentioned. doi 10.1084, 2021 in JEM.
Response:
Thank you for your observations. We have added the reference for the effects of FTY720 on psoriasis as ref 85 and 86. Also, we have added a paragraph regarding the relationship between S1PRs and TRM according to your suggestion. The paragraph has been added at the end of Section 6.
Reviewer 2 Report
This is a non-systematic review of the signaling pathways dependent on the S1P molecule. The review not only includes molecular or basic data, but also the clinical contexts that they imply, fundamentally in pathologies of the skin. The review is understandable, well written and with adequate figures.
Just highlight an error in the numbering of the references (the number is repeated)
Author Response
Reviewer: 2
Comments and suggestions for Authors
This is a non-systematic review of the signaling pathways dependent on the S1P molecule. The review not only includes molecular or basic data, but also the clinical contexts that they imply, fundamentally in pathologies of the skin. The review is understandable, well written and with adequate figures.
Just highlight an error in the numbering of the references (the number is repeated)
Response:
Thank you for your kind review. We have corrected the error in the numbering of the references according to your observation.
Reviewer 3 Report
In this review, the authors talked about the effect of S1P and S1P-S1PR separately and also specifically mentioned both intracellular actions of S1P itself and S1PRs cause cell growth arrest. About this part, I have some concerns,
1. The difference in the mechanism of S1P itself and S1P-S1PR in regulating human keratinocytes/mast cell activities?
Author Response
Reviewer: 3
Comments and suggestions for Authors
In this review, the authors talked about the effect of S1P and S1P-S1PR separately and also specifically mentioned both intracellular actions of S1P itself and S1PRs cause cell growth arrest. About this part, I have some concerns,
The difference in the mechanism of S1P itself and S1P-S1PR in regulating human keratinocytes/mast cell activities?
Response:
Thank you for your observations. S1P causes cell growth arrest intracellularly, and S1PRs cause cell growth arrest via a downstream signaling pathway. We have added a diagram in Figure 3. In mast cells, intracellular S1P causes degranulation and cytokine production by itself. Instead, S1P linked to S1PRs causes effects dependent on the GCPR pathway specific to the receptor of interest. We modified the title of Figure 4.